# MicroRNAs miR-584-5p and miR-425-3p Are Up-Regulated in Plasma of Colorectal Cancer (CRC) Patients: Targeting with Inhibitor Peptide Nucleic Acids Is Associated with Induction of Apoptosis in Colon Cancer Cell Lines

**DOI:** 10.3390/cancers15010128

**Published:** 2022-12-25

**Authors:** Jessica Gasparello, Chiara Papi, Matteo Zurlo, Laura Gambari, Alex Manicardi, Andrea Rozzi, Matteo Ferrarini, Roberto Corradini, Roberto Gambari, Alessia Finotti

**Affiliations:** 1Department of Life Sciences and Biotechnology, University of Ferrara, 44121 Ferrara, Italy; 2Laboratorio Ramses, IRCCS Istituto Ortopedico Rizzoli, 40136 Bologna, Italy; 3Department of Chemistry, Life Sciences and Environmental Sustainability, University of Parma, 43124 Parma, Italy

**Keywords:** colorectal cancer, microRNA, microRNA targeting, peptide nucleic acids, sulforaphane, combo-therapy, liquid biopsy, nutraceuticals

## Abstract

**Simple Summary:**

Starting from a list of nine miRNAs found to be dysregulated in the bloodstream of early-stage colon cancer (CRC) patients, the biological effects of the targeting of the most relevant up-regulated miRNAs with anti-miRNA peptide nucleic acids (PNAs) were verified. The down-regulation of these target miRNAs with PNAs is associated with apoptosis induction in CRC cellular models. Apoptosis induction, which is one of the most important anticancer mechanisms, reaches very high levels when anti-miRNAs PNAs are combined with other pro-apoptotic compounds such as sulforaphane.

**Abstract:**

Liquid biopsy has dramatically changed cancer management in the last decade; however, despite the huge number of miRNA signatures available for diagnostic or prognostic purposes, it is still unclear if dysregulated miRNAs in the bloodstream could be used to develop miRNA-based therapeutic approaches. In one author’s previous work, nine miRNAs were found to be dysregulated in early-stage colon cancer (CRC) patients by NGS analysis followed by RT-dd-PCR validation. In the present study, the biological effects of the targeting of the most relevant dysregulated miRNAs with anti-miRNA peptide nucleic acids (PNAs) were verified, and their anticancer activity in terms of apoptosis induction was evaluated. Our data demonstrate that targeting bloodstream up-regulated miRNAs using anti-miRNA PNAs leads to the down-regulation of target miRNAs associated with inhibition of the activation of the pro-apoptotic pathway in CRC cellular models. Moreover, very high percentages of apoptotic cells were found when the anti-miRNA PNAs were associated with other pro-apoptotic agents, such as sulforaphane (SFN). The presented data sustain the idea that the targeting of miRNAs up-regulated in the bloodstream with a known role in tumor pathology might be a tool for the design of protocols for anti-tumor therapy based on miRNA-targeting molecules.

## 1. Introduction

Liquid biopsy (LB) in cancer is based on the search and analysis of tumor-derived biomarkers in body fluids (e.g., blood and derivatives, saliva, urine, pleural effusions, cerebrospinal fluid, and stool) of tumor-affected patients [1,2,3,4,5,6,7,8,9]. The most commonly investigated tumor-derived entities are circulating tumor DNA (ctDNA), circulating tumor cells (CTCs), circulating microRNAs (c-miRNAs), tumor-derived extracellular vesicles (TD-EVs), and tumor-derived exosomes (TEs) [10,11,12,13,14,15,16]. In the last decade, LB has dramatically changed cancer management, providing two key achievements: the possibility of (a) tracking tumor evolution and (b) obtaining a holistic tumor picture [17,18,19,20,21]. The liquid biopsy approach has demonstrated a high level of heterogeneity of cancer patients (inter-tumor heterogeneity) with respect to the expression of cancer-related biomarkers, including cancer-associated microRNAs [22,23,24]. This might lead to a personalized strategy for the development of anticancer protocols, in the case relevant and differentially expressed miRNAs are targeted.

Cancer-associated miRNA signatures have been reported for gliomas [25], breast cancer [26], melanoma [27], lung cancer [28], hepatocarcinoma [29], gastric cancer [30], and other types of tumors [31,32,33]. Most of them are designed for early diagnostic purposes, while others are used to identify prognostic markers; limited data are available on the possibility that miRNAs found to be dysregulated in the bloodstream may be also therapeutic targets.

We recently reported a distinctive miRNA signature in the blood of colorectal cancer (CRC) patients at surgery, demonstrating the presence of dysregulated miRNAs in early stages of CRC onset and progression [34]. In our study, unlike most previous LB based studies focusing on advanced metastatic CRC, we assessed miRNA dysregulation in blood samples obtained on the day of surgery from patients with primary CRC lesions but no clinical evidence of extra-colonic diffusion. The miRNA profile in plasma isolated from a cohort of 35 CRC patients on the day of surgery was analyzed by next-generation sequencing (NGS) and further confirmed by droplet digital RT-PCR (dd-RT-PCR). The study identified a novel, distinct nine-miRNA signature comprising five up-regulated and four down-regulated miRNAs.

Since the miRNA list provides diagnostic markers as well as possible molecular targets for protocols focusing on “microRNA therapeutics”, the present study was conducted to verify the biological effects of targeting the most relevant dysregulated miRNAs. The objective was to verify whether dysregulated miRNAs in liquid biopsies were at least partially involved in interfering with apoptosis and whether miRNA targeting might lead to the induction of apoptosis in a CRC model system. Moreover, we verified the hypothesis that liquid biopsy is a useful tool for identifying miRNAs to be targeted for the development of new next-generation drugs for cancer treatment.

## 2. Materials and Methods

### 2.1. Materials

The employed chemicals and reagents were all analytical-grade. SFN (D,L-Sulforaphane, 574215-25MG, Merck Millipore, Burlington, MA, USA) was resuspended in DMSO (D8418, Sigma-Aldrich, St. Louis, MO, USA) at 150 mM stock concentration, and single-use aliquots were prepared and stored at −20 °C. At the moment of use, stock aliquots were diluted 1:10 in DMSO.

### 2.2. Synthesis and Characterization of PNAs

Synthesis and characterization of R8-PNA-a15b (H-R_8_-TGTAAACCATGATGTGCT-Gly-NH2) have been reported elsewhere [35] and are fully described in the Appendix A. The identity of the synthetized PNAs was checked with the UPLC-ESI system (see the Appendix A for conditions) and the quantification was performed using the following ε (260 nm) for the nucleobases: Adenine 13,700 M^−1^cm^−1^, Cytosine 6600 M^−1^cm^−1^, Guanine 11700 M^−1^cm^−1^, and Thymine 8600 M^−1^cm^−1^.

R8-PNA-a425: H-R8-GCGGACACGACATTCCCG-Gly-NH2; Yield = 5.54%; UPLC/ESI-MS Rt = 2.59 and 2.95 min, MW calculated: 6172.3 Da; *m/z* found (calculated): 1235.5 (1235.5) [M + 5H]^5+^, 1030.0 (1029.7) [M + 6H]^6+^, 882.8 (882.8) [M + 7H]^7+^, 772.6 (772.5) [M + 8H]^8+^, 686.8 (686.8) [M + 9H]^9+^, 618.3 (618.2) [M + 10H]^10+^ (see Appendix A).

R8-PNA-a584: H-R8-CTCAGTCCCAGGCAAACC-Gly-NH2; Yield = 2% UPLC/ESI-MS Rt = 2.57 min, MW calculated: 6116.3 Da, found: 6117.0 g/mol; *m/z* found (calculated): 1224.3 (1224.3) [M + 5H]^5+^, 1020.4 (1020.4) [M + 6H]^6+^, 874.8 (874.8) [M + 7H]^7+^, 765.6 (765.5) [M + 8H]^8+^, 680.6 (680.6) [M + 9H]^9+^, 612.7 (612.6) [M + 10H]^10+^ (see Appendix A).

PNA-control (unrelated seq): H-R8-ACACTCTACATCACT-Gly-NH2; Yield = 6 % UPLC/ESI-MS Rt = 2.50 min, calculated MW calc. = 5272.5 Da; *m/z* found (calculated): [M + 5H]^5+^, 1055.6 (1055.5) [M-6H]^6+^, 879.5 (879.7) [M + 7H]^7+^, 754.1 (754.2) [M + 8H]^8+^, 660.0 (660.1) [M + 9H]^9+^, 586.5 (586.8) [M + 10H]^10+^ (see Appendix A). Detailed methods are reported in Appendix A.

### 2.3. Cell Culture Conditions

The HT-29 and LoVo cell lines were employed as CRC cellular models [35,36]. Cells were cultured in a humidified atmosphere of 5% CO_2_/air in RPMI 1640 medium (EuroClone, Pero, Milano, Italy), 10% fetal bovine serum (FBS, Biowest, Nuaillé, France), and 100 units/mL penicillin and 100 µg/mL streptomycin (Pen-Strep, Sigma-Aldrich) as described elsewhere [35] (more detailed information on cell culture conditions is provided in Appendix A). Cell growth was monitored using a Z2 Coulter Counter (Coulter Electronics, Hialeah, FL, USA). Cells were seeded at 100.000 cell/well and treated the day after with anti-miRNA PNAs or SFN. A single addition of PNAs and SFN was performed. Both cell lines were exposed to the treatments for 72 h according to the PNA internalization time laps first published by Brognara et al. [36] and previously reported examples of PNA-based treatments [35,37].

### 2.4. RNA Extraction

Cultured cells were detached with trypsin (0,05% trypsin and 0.02% EDTA; Sigma-Aldrich), centrifuged at 1200 rpm for 8 min at 4 °C, washed with DPBS 1X (Gibco, Thermo Fischer Scientific, Waltham, MA, USA), and lysed with Tri-Reagent (Sigma Aldrich, St. Louis, MO, USA), according to the manufacturer’s instructions. The isolated RNA was washed once with cold 75% ethanol, dried, and dissolved in nuclease-free water before use. Obtained RNA was stored at −80 °C until use.

### 2.5. Quantitative Analyses of miRNAs

MicroRNA quantification was performed using real-time RT-qPCR and miRNA-specific primers and probes (reported in Table 1) obtained from Applied Biosystems. Reverse transcriptase (RT) reactions were performed using TaqMan MicroRNA Reverse Transcription Kit (Applied Biosystems, Thermo Fischer Scientific, Waltham, MA, USA) according to the manufacturer’s protocol. All RT reactions, including no-template controls and RT-minus controls, were run in duplicate using TaqMan Universal PCR Master Mix, no AmpErase UNG 2X (Applied Biosystems, Thermo Fischer Scientific, Waltham, MA, USA). Amplifications were performed using the CFX96 Touch Real-Time PCR Detection System (BioRad, Hercules, CA, USA), using the following amplification steps: 95 °C for 10 min, 95 °C for 15 s, and 60 °C for 1 min (for 50 cycles). Data were analyzed using Bio-Rad CFX Manager Software (Bio-Rad, Hercules, CA, USA). The relative expression was calculated using the comparative cycle threshold method, as previously reported [38,39].

### 2.6. Analysis of Apoptosis

Apoptosis was assayed using the Muse Cell Analyzer instrument (Millipore Corporation, Billerica, MA, USA), the Muse Annexin V & Dead Cell Kit and the Muse Caspase-3/7 Kit, as elsewhere reported [35]. Cells were washed with sterile PBS 1X, detached by trypsinization, suspended, and diluted (1:2) with the Muse Annexin V & Dead Cell reagent (see Appendix A). Samples were incubated at room temperature and protected from the light for 15 min, and at the end of incubation, analyzed using the Muse Cell Analyzer and Annexin V and Dead Cell Software Module (Millipore) for data elaboration. Caspase-3/7 activation was detected using the Muse Caspase-3/7 kit, as described elsewhere [35] and extensively described in Appendix A.

### 2.7. Computational Methods

Analysis of drug combination effects was performed using the method developed by Chou and Talalay [40]. The Combination Index (CI) was calculated according to the Chou and Talalay method using the freely available web-based software compuSyn (www.combosyn.com (accessed on 12 December 2022) for drug synergy analysis. The interaction between drugs was considered synergic for a CI lower than 1, and CI values close to 1 were considered indicative of an additive effect, while values of more than 1 indicated antagonism.

## 3. Results

### 3.1. MicroRNAs miR-425-3p, miR-584-5p, and miR-15b-5p Are Frequently Up-Regulated in CRC Patients

We have comparatively assessed the up-regulation of the nine-miRNA signature previously reported by some of us [34] and found that three microRNAs were up-regulated in a high number of CRC patients. For instance, and in agreement with a previously reported study [34], miR-15b-5p was found up-regulated in 9 patients and down-regulated in 1 patient, while in 25 CRC patients its expression was found to be similar to that of control tumor-free individuals (Figure 1A). Another 2 microRNAs, miR-584-5p and miR-425-3p, were found to be up-regulated in 25 and 12 CRC patients, respectively. Of relevance, miR-425 was not found to be down-regulated in the cohorts of patients described by the same authors of the present work [34].

In order to perform comparisons, for each miRNA belonging to the nine-miRNA signature, we applied the following algorithm to calculate the “CRC-index”: (number of patients displaying up-regulation–number of patients displaying down-regulation)/total number of patients analyzed. We compared the data obtained (Figure 1B) with the data of the well-studied miR-221-3p, which has been firmly established as an oncomiRNA up-regulated in several tumor tissues and liquid biopsies, such as those from hepatocarcinoma, as reported in the following studies: Tan et al. [41], pancreatic cancer; Li et al. [42], ovarian cancer; Xie et al. [43], osteosarcoma; Gong et al. [44], lung cancer; Zhou et al. [45] and Visani et al. [46], glioblastoma. Of relevance for our study, miR-221 is up-regulated in colorectal carcinomas [47,48,49,50,51].

When this analysis was performed, three miRNAs were found to be up-regulated in a number of CRC patients at a rate higher (miR-584-5p and miR-425-3p) or similar (miR-15b-5p) to miR-221. The CRC indexes were 0.7, 0.3, and 0.2 for miR-584-5p, miR-425-3p and miR-15b-5p, respectively, while it was found to be 0.1 for miR-221-3p (Figure 1B). Accordingly, we focused on miR-584-5p and miR-425-3p, while miR-15b-5p was used as control, since we have already characterized this microRNA on the basis of several reports describing this miRNA as an oncomiRNA up-regulated in CRC samples [35,52,53,54,55]. In these studies, miR-15b-5p was found to be up-regulated in CRC patients compared to healthy controls not only in tumor biopsies, as reported by Xi et al. [52], but also in liquid biopsies [53]. Moreover, miR-15b-5p inhibition was shown to have an important therapeutic function in reducing tumor migration and metastasis [54,55] and inducing apoptosis [35].

### 3.2. Selective Targeting of miR-584-5p and miR-425-3p with R8-PNA-a584 and R8-PNA-a425

High expression of miR-584-5 and miR-425-3p was confirmed in colon cancer HT-29 and LoVo cells by dd-RT-qPCR (Appendix A).

In order to determine the functions of these microRNAs, we determined the effects of their inhibition using, as down-regulating molecules, peptide nucleic acids (PNAs) functionalized with an R8 peptide, in order to maximize their uptake by target cells as previously reported [35,36,37,38]. The PNA-mediated down-regulation of miR-584-5p and miR-425-3p is a novel strategy in cancer research and treatment, while PNA-mediated inhibition of miR-15b-5p has been reported previously, but only using the HT-29 cell line [35].

Targeting of miR-584-5p and miR-425-3p with R8-PNA-a425 and R8-PNA-a584 caused a selective down-regulation of the target miRNA (Figure 2A). This was expected since anti-miRNA PNAs have been reported to retain the specificity of action against complementary miRNA sequences, as reported in studies published elsewhere [56,57,58]. As expected, R8-PNA-a425 caused inhibition of miR-425-3p but only minor effects on other miRNAs, such as miR-584-5p, miR-15b-5p, and miR-210-3p (Figure 2B). Conversely, R8-PNA-a584 caused inhibition of miR-584-5p but only minor effects on other miRNAs, such as miR-425-3p, miR-15b-5p, and miR-210-3p. The specificity of R8-PNA-a15b has been discussed elsewhere [35] and was confirmed by the control experiment shown in Figure 2C.

Interestingly, all these PNAs caused a reproducible slight inhibition of cell growth of HT-29 cells (Figure 2D). No effects of the R8 peptide on cell growth and apoptosis were expected, as discussed in reported observations by our and other research groups [35,59,60,61,62]. The results obtained using R8-PNA-a425 and R8-PNA-a584 demonstrate that together with the reduction in specific hybridization (Figure 2A–C) to the target miRNA sequences, a statistically significant inhibition of cell growth was detected (Figure 2D). These novel data (to our knowledge, no study is available on the effects of PNA against miR-584-5p and miR-425-3p) support the idea that inhibition of these miRNAs in colon cancer cells is associated with anti-tumor activity in vitro.

### 3.3. Targeting of miR-584-5p and miR-425-3p with R8-PNA-a584 and R8-PNA-a425 Is Associated with Weak Pro-Apoptotic Effects

When PNAs targeting miR-584-5p and miR-425-3p were singularly administered to HT-29 cells, a weak but highly reproducible increase in the apoptotic cell rate was observed, employing the Annexin-V assay. In the representative experiment shown in Figure 3A, a summary, including three independent experiments, is presented. The increase was in the order of 10% (with the apoptotic rate of control samples subtracted) when 8 µM R8-PNA-a584 or R8-PNA-a425 was used. In panel B of Figure 3, exemplificative plots of the apoptotic profile are provided. A control unrelated-sequence PNA was found inactive in reducing intracellular levels of target miRNAs and consequently, in inducing pro-apoptotic effects (Appendix A). Apoptotic induction is associated with a clear change in the morphological features of treated cells (Figure 3C).

### 3.4. Simultaneous Targeting of miR-584-5p and miR-425-3p or miR-584-5p, miR-425-3p, and miR-15b-5p Is Associated with Synergic Pro-Apoptotic Effects in CRC Cellular Models

Since the induction of apoptosis is considered among the most interesting anti-tumor strategies [63], we verified whether the herein-investigated R8-PNA-a584 and R8-PNA-a425 might exhibit synergistic effects (a) if employed in combination, or (b) if combined with the elsewhere-investigated R8-PNA-a15b [35].

To this aim, HT-29 (Figure 4A–C) and LoVo (Figure 4D–F) cells were cultured for 72 h in the presence of (a) suboptimal concentrations (4 µM) of singularly administrated anti-miRNA PNAs, or (b) a combination of 2 or 3 PNAs. The effects of single PNAs or PNA combinations on target miRNAs and on other unrelated miRNAs were verified (Appendix A), confirming the target specificity of the employed PNAs. Simultaneous miRNA targeting using two-PNA combinations led to additive (R8-PNA-a584 plus R8-PNA-a425) or synergistic (R8-PNA-a584 plus R8-PNA-a15b; R8-PNA-a425 plus R8-PNA-a15b) effects when suboptimal concentrations of PNAs were employed. This was found using either the Annexin V (Figure 4A,D) or the Caspase 3/7 (Figure 4B,E) assays. Interestingly, the triple combination of R8-PNA-a584, R8-PNA-a425, and R8-PNA-a15b led to the maximum level of apoptosis as assessed by Annexin V and Caspase 3/7 assay. Representative key apoptotic profiles are reported in Figure 4C,F. A complete set of representative apoptotic profiles can be found in the Appendix A. Remarkably, a control unrelated scrambled PNA used at a concentration of 12 μM (corresponding to the sum of the three investigated PNAs used at 4 μM when singularly administered) was found to have very low effects on apoptosis induction (Appendix A). In fact, the increase in the apoptotic rate with respect to the control samples was found to be lower than 10% and 5% in the Annexin V and Caspase 3/7 assays, respectively.

Altogether, our data demonstrate that high levels of apoptosis induction can be reached in combined treatments using, even at suboptimal concentrations, PNAs targeting miRNAs found to be dysregulated in the bloodstream of CRC patients.

### 3.5. Co-Treatment of CRC-Cells with Sulforaphane and R8-PNA-a425 or R8-PNA-a584: Effects on Apoptosis

In order to verify the possibility of inducing a high level of apoptosis by targeting miRNAs dysregulated in the “9-miRNAsignature”, another strategy was assessed, in which the anti-miRNA PNAs employed in this study were combined with a well-known pro-apoptotic compound: sulforaphane (SFN) [35,64,65,66]. SFN was shown to induce apoptosis in both CRC cellular models (Appendix A), and according to the apoptotic profile shown in Appendix A, a SFN concentration of 30 µM was selected to set up combinations with anti-miRNA PNAs.

When singular treatments were compared with combined treatments using R8-PNA-a584, R8-PNA-a425, and SFN, the rate of apoptosis induction obtained using the combined approach was significantly higher. The employed concentrations of PNAs and SFN were based on previously reported observations by our and other groups [35,66,67]. The treatments were carried out for 72 h. Figure 5 reports the data obtained by combining SFN with R8-PNA-a584 (Figure 5A,B,E,F) and R8-PNA-a425 (Figure 5C,D,G,H). In both cases, high rates of apoptotic cells (over 75%) were obtained, when the Annexin V assay was performed. The means of three independent experiments are reported in Figure 5A,C,E,G, while representative plots are presented in Figure 5B,D,F,H. Synergistic effects of the combined treatments (SFN + R8-PNA-a425; SFN + R8-PNA-a584) were confirmed after analyzing the data considering the increase in the proportion of apoptotic cells with respect to control untreated (or DMSO-treated) cells. Representative raw data can be found in the Appendix A. No synergic effects were observed when SFN was added to an unrelated sequence PNA (Appendix A).

## 4. Discussion

Colon cancer (CRC) patients frequently express high levels of miR-425-3p and miR-584-5p, which are (together with miR-15b-5p) the most frequently up-regulated microRNAs among those belonging to the nine-miRNA CRC-associated signature reported by Gasparello et al. [34]. The oncogenic or tumor-suppressing role of the two microRNAs is under debate. Information supporting that these miRNAs are associated with CRC is lacking, and their biological function is still not known in detail. Literature data describe miR-584-5p as a tumor suppressor miRNA in different cancer types including gastric cancer [68] and lung cancer [69,70]. On the contrary, fluctuating expression of miR-584 in primary and high-grade gastric cancer was recently proposed [71]. In that study, the expression level of miR-584 was studied in primary gastric cancers versus healthy control gastric mucosa samples using the RT-qPCR method.

The conclusion of that study was that miR-584 undergoes up-regulation in the stage of primary tumor formation; however, it becomes down-regulated upon the progression of gastric cancer. These findings suggest the potential of miR-584 as a diagnostic or prognostic biomarker in gastric cancer.

Regarding miR-425, this microRNA is associated with poor prognosis and promotes cancer cell progression by targeting PTEN in breast cancer [72] and Dickkopf 3 in bladder cancer [73]. MicroRNA miR-425 dysregulation is also associated with prostate [74] and lung [75] cancers.

In this paper, we report for the first time, to the best of our knowledge, that targeting of miR-584 and miR-425 using PNAs as anti-miRNA molecules leads to down-regulation of target miRNAs in colon cancer cells, which is associated with inhibition of in vitro cell growth and activation of the pro-apoptotic pathway. The highest efficiency was obtained when the simultaneous targeting of two or three miRNAs was performed, according to the data shown in Figure 4. Moreover, in our case, combining more anti-miRNA PNAs allowed us to use suboptimal concentrations of each PNA, reducing any risk of aspecific targeting. Therefore, this strategy is expected to reduce the side effects of single anti-tumor PNAs.

The second conclusion of our paper is that R8-PNA-a584 and R8-PNA-a425 might be associated with other pro-apoptotic agents as therapeutic tools. Recently, the interest in the search for nutraceutical compounds able to exert therapeutic effects on cancer and other chronic diseases has rapidly increased. Among them, organosulfur compounds seem to be a promising class of compounds [76,77]. Here, we focused on SFN, an organosulfur compound derived from edible plants of the *Brassica* genera, which have been largely studied in the field of cancer due to their known pro-apoptotic effects on hepatocarcinoma [78], prostate cancer [79], and bladder cancer [80] cells. With respect to colorectal cancer, SFN is known to inhibit phase I metabolic enzymes, induce phase II metabolic enzymes, and induce apoptosis, cell cycle arrest, autophagy, and inhibition of tumor angiogenesis [81]. To our knowledge, few reports are available in the literature on a combination of SFN and miRNA-targeting molecules, and no report is available on the effects of combining SFN administration with inhibition of miR-584 and miR-425. Our data demonstrate that this combined treatment leads to a very high proportion of apoptotic HT-29 cells (over 75% for both combinations: R8-PNA-a584 plus SFN and R8-PNA-a425 plus SFN).

In order to determine possible synergisms between R8-PNA-a584, R8-PNA-a425, R8-RNA-a15b and SFN, we applied the Chou–Talalay method [40], which does not require knowledge of the mechanisms of action of each drug. In this method, a Combination Index (CI) < 1 indicates synergisms, a CI = 1 indicates additivity, and a CI > 1 indicates antagonisms [82]. In particular, co-treatment of different R8-PNAs showed in the majority of case synergism. For instance, the CI for the co-treatment of R8-PNA-a15b and R8-PNA-a425 was found to be 0.52 (Figure 4A); the combination of R8-PNA-a15b and R8-PNA-a584 led to a score of 0.73 (Figure 4A); finally, the triple-treatment R8-PNA-a15b plus R8-PNA-a584 and R8-PNA-a425 led to a score of 0.69 (Figure 4A). Similarly, co-treatment of R8-PNAs with SFN showed synergism. When SFN was used in combined treatment, the CI was 0.44 when R8-PNA-a4584 was used (Figure 5A) and 0.40 when R8-PNA-a425 was used (Figure 5C). The CI values in the different experiments performed are reported in Appendix A.

Our data verify that targeting up-regulated miRNAs with a known involvement in tumor pathology might be useful for designing selective protocols for anti-tumor therapy. In consideration of the high patient-to-patient variability of miRNA up-regulation, this approach might be considered in precision medicine for personalized treatments of cancer patients.

Further in vitro studies will clarify the effects of these PNAs on the molecular targets of miR-15b-5p, miR-425-3p, and miR-584-5p (Appendix A). Considering the fact that the number of these targets is very high, transcriptomic and proteomic analyses are necessary to obtain a comprehensive view of the effects of PNA-based treatments. These studies will also help in clarifying the differences between the effects of PNA and SFN treatment on HT-29 and LoVo cells. Differences are expected, as these cell lines are different with respect to origin and genomic/epigenomic characteristics. HT-29 cells are derived from a primary tumor, while LoVo cells come from a late-stage (Dukes C, grade V) metastatic tumor. HT-29 cells carry a WT KRAS profile, while LoVo cells harbor the G13D KRAS mutation [83]. Another difference is the microsatellite stability; in fact, as reported by Yashiro et al., LoVo has an MSI phenotype, while HT-29 cells do not [84]. Some of these features may influence the response to SFN treatments, as in the case of other phytochemical compounds such as curcumin, the activity of which may be affected by the microsatellite status [85].

Moreover, the validation of the efficacy of co-treatments using different PNAs or the combined strategy based on anti-miRNA PNA and SFN is a required step for proposing this approach for the possible development of tailored protocols for therapeutic interventions in the management of colon cancer patients. In vivo experiments are feasible, as demonstrated several studies showing in vivo activity of SFN [86,87,88]. Concerning PNAs, examples of the employment of these reagents in in vivo tumor models have been reported [89,90].

As a final consideration, this manuscript provides a proof of concept that screening with liquid biopsy can be performed to identify dysregulated miRNAs, which can be targeted with the PNA approach to obtain pro-apoptotic effects on cancer cells. This has great potential value, and can offer perspectives on personalized treatments for precision medicine. Moreover, the presented technology can be translated to other diseases, such as chronic musculoskeletal pathologies, where a modulation of apoptosis can be of therapeutical value [91,92].

## 5. Conclusions

In conclusion, liquid biopsy, as a valid tool for diagnostic and prognostic purposes, might be a useful approach to identifying molecular entities (e.g., DNA mutations, dysregulated miRNAs), which may be druggable targets for CRC treatment.

Our results show that the anti-miRNA strategy could lead to relevant therapeutic inhibition of miRNA-dependent effects and that PNA-based anti-miRNA molecules are very promising reagents for regulating tumor cell growth; further research on PNA analogues to increase the efficiency of delivery, stability, and control of intracellular distribution for specific targets, i.e., mature miRNA, pre-miRNA, or pri-miRNA, are further steps for the selection of the best candidate drugs. The use of anti-miRNA PNAs should be limited to specific tumor types (for which an association with the target miRNAs is demonstrated) and selected patients (carrying tumors exhibiting up-regulation of the target miRNAs when comparison with non-tumor tissue samples is available).

Finally, our study strongly indicates that the combined treatment of target cells with a) PNAs targeting multiple miRNAs (in this study, PNAs targeting miR-425-3p, miR-584-5p, miR-15b-5p) or b) combinations of anti-miRNA PNAs and anti-tumor agents (in this study, SFN) is a promising strategy to increase efficacy while limiting, at least in theory, side effects.

## Figures and Tables

**Figure 1 cancers-15-00128-f001:**
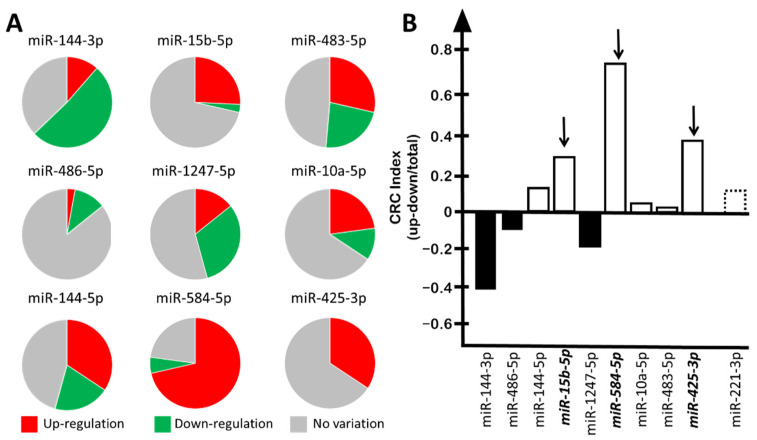
Content of the miRNAs belonging to the “9-miRNA signature” in the plasma from CRC patients. (**A**) Pie chart reporting the number of CRC patients in which each indicated miRNA was found up-regulated (red), down-regulated (green), or to have no substantial variation detected (gray). (**B**) Summary of the “CRC-index” values of the “9-miRNA signature” compared to miR-221-3p. “CRC-index”: (number of patients displaying up-regulation–number of patients displaying down-regulation)/total number of patients analyzed. Up-regulated miRNAs are reported as white boxes, and down-regulated miRNAs as black boxes. Dotted box: miR-221-3p.

**Figure 2 cancers-15-00128-f002:**
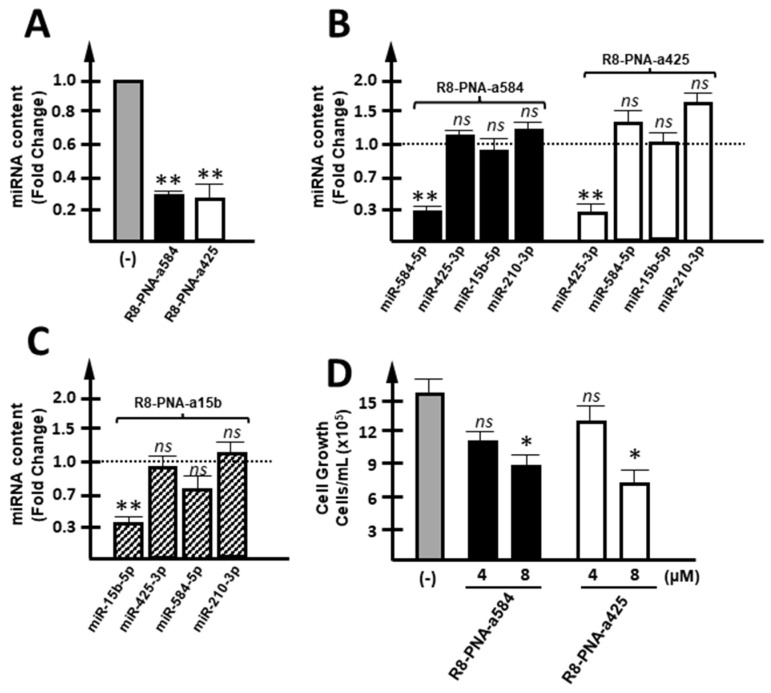
Effects of R8-PNA-a425 and R8-PNA-a584 on miRNA content. (**A**) Effects of anti-miRNA PNAs on their respective target miRNA, after performing 72 h of culturing of HT-29 cells. (**B**) Possible off-target effects were verified for both R8-PNA-a584 (white boxes) and R8-PNA-a425 (black boxes): miR-425-3p, miR-15b-5p, and miR-210-3p content was calculated for R8-PNA-a584-treated HT-29 cells, while miR-584-5p, miR-15b-5p, and miR-210-3p were quantified when R8-PNA-a425 was employed. (**C**) Specificity of R8-PNA-a15 was confirmed by quantifying miR-15b-5p, miR-425-3p, miR-584-5p, and miR-210-3p content in HT-29 R8-PNA-a15b-treated cells. (**D**) Effects on HT-29 cell growth of 4 and 8 µM R8-PNA-a584 (black boxes), R8-PNA-a425 (white boxes). (* *p* < 0.05, ** *p* < 0.01, ns: not significant, n = 3 independent experiments).

**Figure 3 cancers-15-00128-f003:**
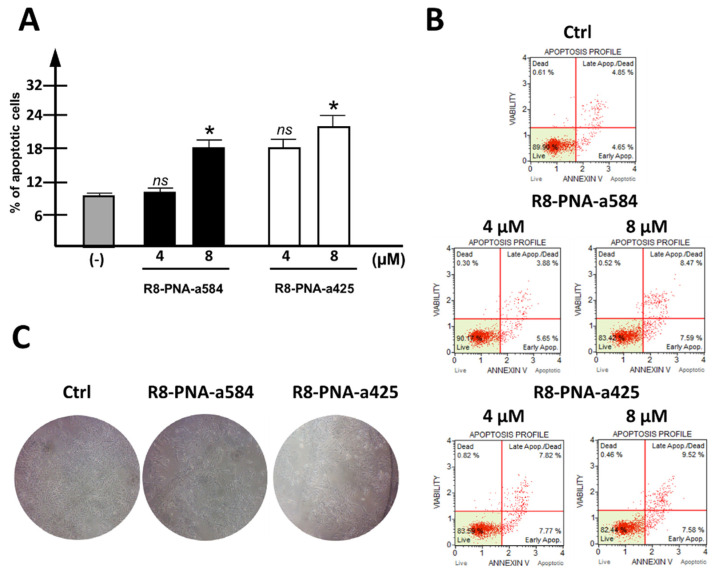
Effects of R8-PNA-a425 and R8-PNA-a584 on apoptosis. (**A**) Effects of anti-miRNA PNAs on apoptosis induction in HT-29 cells treated for 72 h with 4 or 8 µM R8-PNA-a584 (black boxes) or R8-PNA-a425 (white boxes). (**B**) Representative plots of the apoptotic profile are reported. (**C**) Representative pictures showing the morphological appearance of HT-29-treated cells are reported. (* *p* < 0.05, ns: not significant; n = 3 independent experiments).

**Figure 4 cancers-15-00128-f004:**
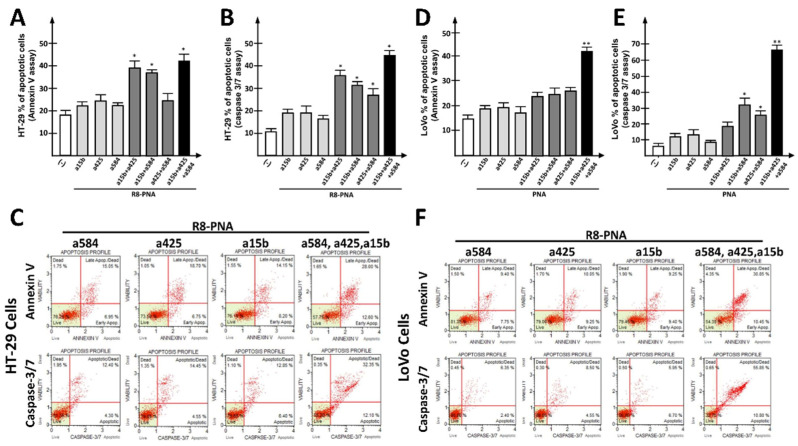
Effects of PNA combinations on apoptosis induction. (**A**,**D**) Effects of combinations of anti-miRNA PNAs on apoptosis induction in HT-29 (**A**) and LoVo (**D**) cells treated for 72 h with a 4 µM concentration of each PNA, as determined using Annexin V assay. Singularly administrated PNA (light gray boxes), combinations of two PNAs (dark gray boxes), triple-PNA combination (black box). The apoptotic profile was determined using Annexin V assay. (**B**,**E**) Effects of anti-miRNA PNA combinations on apoptosis induction in HT-29 (**B**) or LoVo (**E**) cells treated for 72 h with a 4 µM concentration of each PNA, as determined using Caspase 3/7 assay. Singularly administrated PNA (light gray boxes), combinations of two PNAs (dark gray boxes), triple-PNA combination (black box). (**C**,**F**) Representative plots obtained by Annexin V (upper line) and Caspase 3/7 (lower line) assays in HT-29 (**C**) or LoVo (**F**) cells. (* *p* < 0.05, ** *p* < 0.01, double-PNA treatment was compared with singularly administrated PNA, while triple-PNA treatment was compared with double-PNA combinations. n = 3 independent experiments).

**Figure 5 cancers-15-00128-f005:**
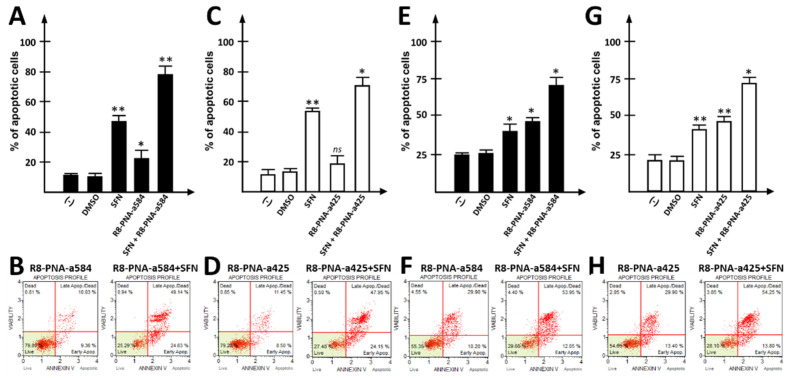
Effects of anti-miRNA PNAs combined with sulforaphane (SFN) on apoptosis induction. (**A**,**E**) Effects of anti-miRNA PNA against miR-584 (R8-PNA-a584) combined with active SFN on apoptosis induction in HT-29 (**A**) and LoVo (**E**) cells treated for 72 h. Apoptosis profile was determined using Annexin V assay. (**B**,**F**) Representative Annexin V assay plots. (**C**,**G**) Effects of anti-miRNA PNA against miR-425 (R8-PNA-a425) combined with active SFN concentration on apoptosis induction in HT-29 (**C**) and LoVo (**G**) cells. Apoptosis profile was determined using Annexin V assay. (**D**,**H**) Representative plots obtained by Annexin V assays. (* *p* < 0.05, ** *p* < 0.01, ns: not significant, n = 3 independent experiments).

**Table 1 cancers-15-00128-t001:** Complete list of the employed miRNA-TaqMan assays.

miRNA Name	Assay ID
hsa-miR-15b-5p	000390
hsa-miR-210-3p	000512
hsa-miR-425-3p	002302
hsa-miR-584-5p	001624
hsa-snRNA U6	001973
hsa-let-7c-5p	000379

## Data Availability

All the data are included in the main text and Appendix A. Further details and data access will be made freely available by the corresponding authors upon reasonable request.

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
