# Peer review of "MicroRNAs miR-584-5p and miR-425-3p Are Up-Regulated in Plasma of Colorectal Cancer (CRC) Patients: Targeting with Inhibitor Peptide Nucleic Acids Is Associated with Induction of Apoptosis in Colon Cancer Cell Lines"

_cancers, 2022, doi:10.3390/cancers15010128_

Round 1
Reviewer 1 Report
The manuscript aims to explore the potential therapy of antagonist of miR-425 and miR-584 in colon cancer treatment based on their previous findings. Authors synthesized several peptide nucleic acid (PNA) antagonists against those two miRNAs, and found these PNAs can efficiently induced apoptosis in colon cancer cells HT-29 and LoVo. In addition, these two PNAs exhibit synergistic effect for inducing apoptosis when combined with sulforaphane (SFN). However, the data doesn't fully support the conclusion and need to be improved.
1. In line183, results about miR15-3p is a previous research of authors, which should be showing the reference; In line187, authors cited one of their research, which should be stated clearly and shown as a self citation.
2. miRNAs were well known to regulate the expression of mRNA, whether the target genes were altered correspondingly upon the treatment of PNAs? The target genes should be detected as well.
3. Author claimed that there is synergic effect for two PNA panel, Three PNA panel, and anti-miR-584 PNA or anti-miR-425 PNA combined with SFN, but it is lacking evidence, the calculation for combination index or synergistic score should be revealed.
4. It looks like the performances of HT-29 and LoVo upon PNAs and SFN treatments are different, whether this is due to the differential expression of target genes or microsatellite stability?
5. In line356, authors quoted a research related to miR-584, but it actually is about miR-548, the right reference should be PMC7345521.
Author Response
Reviewer #1.
Point 1. In line 183, results about miR15-3p is a previous research of authors, which should be showing the reference.
Answer. We thank for the comments and we made a citation of our previous research study (reference [34]). In addition, in order to clarify the novelty of the PNA targeting the three miRNAs described in the present study, the following sentence has been also included: “The PNA-mediated down-regulation of miR-584-5p and miR-425-3p is a novel strategy in cancer research and treatment, while PNA-mediated inhibition of miR-15b-5p has been reported previously, but using only the HT-29 cell line [35]” (line 230-233)
Point 2. In line 187, authors cited one of their research, which should be stated clearly and shown as a self citation.
Answer. We have extensively modified the sentence located at page 4 (lines 185-189) as follows: “We have comparatively assessed the upregulation of the nine-miRNA signature previously reported by some of us [34] and found that three microRNAs are upregulated in the highest number of CRC patients. For instance, and in agreement with the previously reported study [34], miR-15b-5p was found up-regulated in 9 patients, down-regulated in 1 patient, while in 25 CRC patients its expression was found similar to control tumor-free individuals (Figure 1A)”. This makes clear that reference [34] is a previous study of our research team that has been extended and applied to miRNA therapeutics by the present study.
Point 3. miRNAs were well known to regulate the expression of mRNA, whether the target genes were altered correspondingly upon the treatment of PNAs? The target genes should be detected as well.
Answer. The supplementary tables Table S1, Table S2 and S3 were added as supplementary materials, including the list of predicted and validated targets of the three miRNAs here studied/targeted. This clarifies the important research issue raised by the reviewer, i.e. identification of the target of the miRNAs pharmacologically regulated by PNAs. Despite the fact that this was not the major point of the paper, we agree that this point should be discussed. This was done by the sentence “Further in vitro studies will clarify the effects of these PNAs on the molecular targets of miR-15b-5p, miR-425-3p and miR-584-5p. Considering the fact that the number of these targets is very high, transcriptomic and proteomic analyses will be necessary to obtain a comprehensive view that the effects of PNA-based treatments.” (page 11, lines 417-420).
Point 4. Author claimed that there is synergic effect for two PNA panel, Three PNA panel, and anti-miR-584 PNA or anti-miR-425 PNA combined with SFN, but it is lacking evidence, the calculation for combination index or synergistic score should be revealed.
Answer. Thank you for this suggestion. We applied the Chou-Talalay algorithm, which was used by many other research groups to identify synergist effects of pharmacological treatment. The algorithm is described in the Material and methods section (2.7) and examples of the CI indexes obtained have been discussed by including the sentence “In order to determine possible synergisms between PNAs and PNAs and SFN, we applied the Chou-Talalay method [40], that does not require …… treatment the CI was 0.44 when PNA-a4584 was used (Figure 5A) and 0.40 when PNA-a425 was used (Figure 5C). Examples of CI values in the different experiments performed are reported in Supplementary Table 4” (page 10, lines 401-407; page 11 lines 408-411). Information on calculated CI, are available in Supplementary materials, Supplementary Table S4.
Point 5. It looks like the performances of HT-29 and LoVo upon PNAs and SFN treatments are different, whether this is due to the differential expression of target genes or microsatellite stability?
Answer. We do not know the effects of the treatment on target genes (see also Point 3). However, we have discussed the differences between HT-29 and LoVo at page 11, lines 420-430 by including the sentence “These studies will also help in clarifying the difference obtained when the effects of PNAs and SFN treatments are compared in HT-29 and cells. Differences are not unexpected, as these cell lines are different with respect to origin and genomic/epigenomic characteristics …….. the activity of which may be affected by the microsatellite status [86]”.
Point 6. In line 356, authors quoted a research related to miR-584, but it actually is about miR-548, the right reference should be PMC7345521.
Answer. We checked the reference list quoted at the beginning of the discussion section, verifying that the reference PMC7345521 was properly cited in the sentence “On the contrary, fluctuating expression of miR-584 in primary and high-grade gastric cancer was recently proposed [72]”.
Sincerely,
Alessia Finotti
Department of Life Sciences and Biotechnology, University of Ferrara
Reviewer 2 Report
The article by Finotti et al. demonstrated two microRNAs associated with in vivo cell growth and activation of the pro-apoptotic pathway - miR-425 and miR-584, while also proposed R8-PNA-a584 and R8-PNA-a425 as two pro-apoptotic agents that potentially can be used for cancer therapy. Overall, the findings made in this article are high in novelty. However, I suggest a major revision of the presentation about the results.
Minor comments:
1) The authors need to be consistent in the written formats of the PNAs. For example, in line 224 under results 3.2, the authors claimed that these PNAs were functionalized with an R8 peptide, which is why they were referred to as "R8-PNA-a425" or "R8-PNA-a584" most of the time throughout the article. However, the authors also used "PNA-a584" in line 238 and several other lines, which could be confusing and misleading to readers.
2) Color scheme also needs to be consistent throughout the figures. For example, in Figures 2D, 3, and 5, when only the "R8-PNA-a425" and "R8-PNA-a584" groups were presented, authors used white-colored bars for R8-PNA-a425 treated groups and black-colored bars for R8-PNA-a584 treated groups. In Figure 2B, however, the colors are opposite, which could be confusing and misleading to readers.
3) Line 290 refers to Figure S4, which I think is a wrong reference.
Major comments:
1) Figure 2B needs to include the miRNA content of targeted micro RNAs (miR-584-5p and miR-425-3p and the statistical results showing that the PNA treatments significantly reduce them), just like the format used in Figure 2C. I see this data in supplementary Figure S6; however, I suggest adding it to the manuscript figure.
2) Data in Figure 3A - how long after the administration of PNAs when this data collected? This is not mentioned in the methods part and needs to be justified. Is there a time-course preliminary study done to help the authors decide the optimal treatment time? If so, a reference to the preliminary study is required.
3) Figure 3A - no statistical analysis or no statistical significance among the treatment groups?
4) Compare data presented in Figure 3A and Figure S5, 8uM PNA-a584 has a ~9% increase of apoptotic cells, while 8 uM un-related PNA sequence has a ~6% increase. And the authors claim "a control unrelated sequence PNAs was found inactive in inducing pro-apoptotic effects" in lines 260-262.
I think if the unrelated sequence PNA treatment is "inactive," then the same criteria should be applied to PNA-a584 treatment, meaning that the authors cannot say PNA-a584 treatment has pro-apoptotic effects. This part needs to be justified if the authors want to keep that conclusion.
5) Data presented in Figure 2C, 2B-"PNA-a584", and Figure S7A are consistent. However, the data presented in Figure 2B-"PNA-a425" and Figure S7A are inconsistent. For example, the microRNA content of miR-15b-5p after PNA-a425 treatment in Figure 2B is 1.0 FC, and in Figure S7A is 0.6 FC. Same cell line (HT-29), how can the results be off so much? Are these two figures based on two different sets of data? Why they are not combined and averaged? This part needs to be justified.
6) Line 291-292, where is the data that supports this statement? A figure reference should be added here.
7) In Figure S7, the figure legend mentions C and D panels, but they are missing in the figure.
8) In Figure S9, the figure legend is missing panels C and D.
9) The data presented in Figure 5A and Figure S10A are inconsistent. Same cell line (HT29) and the same Annexin V assay, in Figure 5A, the SFN group has a ~ 40% increase of apoptotic cells compared to the control (untreated, -), while in Figure S10A the SFN group has a ~ 20% increase of apoptotic cells, assuming it is also comparing to the control. How can the results be off so much? Are these two figures based on two different sets of data? Why they are not combined and averaged? This part needs to be justified.
10) From Figure 3 to Figure 5, there is no indication of statistical analysis or statistical significance among the treatment groups. This greatly affects the significance of the study content and the authors should include a justification in the discussion indicating the limits of the study as well as reasoning the scientific soundness of their findings.
Author Response
Reviewer #2.
Point 1. The authors need to be consistent in the written formats of the PNAs. For example, in line 224 under results 3.2, the authors claimed that these PNAs were functionalized with an R8 peptide, which is why they were referred to as "R8-PNA-a425" or "R8-PNA-a584" most of the time throughout the article. However, the authors also used "PNA-a584" in line 238 and several other lines, which could be confusing and misleading to readers.
Answer. The same nomenclature of each PNA (R8-PNA-a15b, R8-PNA-a425 and R8-PNA-a584) was used throughout the text and within the figures.
Point 2. Color scheme also needs to be consistent throughout the figures. For example, in Figures 2D, 3, and 5, when only the "R8-PNA-a425" and "R8-PNA-a584" groups were presented, authors used white-colored bars for R8-PNA-a425 treated groups and black-colored bars for R8-PNA-a584 treated groups. In Figure 2B, however, the colors are opposite, which could be confusing and misleading to readers.
Answer. This is an important style issue. In all the figures the same colour was used for symbols/histogram referring to each PNA; for instance, R8-PNA-a425 was always “white” coloured, while R8-PNA-a584 was “black” coloured; the PNA-a15b was coloured with black and white lines. Please, note that the figures were amended by adding ‘R8-’ before PNA-a15b, PNA-a425 and PNA-a584, in order to be consistent with the PNA nomenclature used along the text.
Point 3. Line 290 refers to Figure S4, which I think is a wrong reference.
Answer. The correct supplementary figure (S5) was cited in the revised version of the manuscript.
Point 4. Figure 2B needs to include the miRNA content of targeted micro RNAs (miR-584-5p and miR-425-3p and the statistical results showing that the PNA treatments significantly reduce them), just like the format used in Figure 2C. I see this data in supplementary Figure S6; however, I suggest adding it to the manuscript figure.
Answer. The content of targeted miRNAs was added in panel B of Figure 2
Point 5. Data in Figure 3A - how long after the administration of PNAs when this data collected? This is not mentioned in the methods part and needs to be justified. Is there a time-course preliminary study done to help the authors decide the optimal treatment time? If so, a reference to the preliminary study is required.
Answer. Data were collected 72 hours after the treatment with PNAs, according to the PNA internalization time laps, reported by Brognara et al [36]. The time of PNA administration was added in materials and methods and the new reference [36] was added. In order to better explain how PNAs were administered, the following sentence was added: “Cells were seeded at 100.000 cell/well and treated the day after with PNAs or SFN. A single addition of PNAs and SFN was performed. Both cell lines were exposed to the treatments for 72 h according to the PNA internalization time laps first published by Brognara et al. [36] and previously reported examples of PNA-based treatments [35,37].” (page 3, lines 127-131).
Point 6. Figure 3A - no statistical analysis or no statistical significance among the treatment groups?
Answer. Statistical analysis was added to Figure 3.
Point 7. Compare data presented in Figure 3A and Figure S5, 8uM PNA-a584 has a ~9% increase of apoptotic cells, while 8 uM un-related PNA sequence has a ~6% increase. And the authors claim "a control unrelated sequence PNAs was found inactive in inducing pro-apoptotic effects" in lines 260-262. I think if the unrelated sequence PNA treatment is "inactive," then the same criteria should be applied to PNA-a584 treatment, meaning that the authors cannot say PNA-a584 treatment has pro-apoptotic effects. This part needs to be justified if the authors want to keep that conclusion.
Answer. We confirm (as stated in the Title of section 3.3, that targeting of miR-584-5p and miR-425-3p with R8-PNA-a584 and R8-PNA-a425 is associated with weak pro-apoptotic effects. This is the reason to consider them, in any case, in combined treatments. We also verified that the control untreated cells exhibit an endogenous pro-apoptotic activity. However, after subtraction of the background levels of apoptosis (in control untreated cells), the PNA-treated samples were always (even slightly) more active than unrelated PNA control.
Point 8. Data presented in Figure 2C, 2B-"PNA-a584", and Figure S7A are consistent. However, the data presented in Figure 2B-"PNA-a425" and Figure S7A are inconsistent. For example, the microRNA content of miR-15b-5p after PNA-a425 treatment in Figure 2B is 1.0 FC, and in Figure S7A is 0.6 FC. Same cell line (HT-29), how can the results be off so much? Are these two figures based on two different sets of data? Why they are not combined and averaged? This part needs to be justified.
Answer. Figure are obtained analysing two different sets of data: one set (3 independent experiments) was performed for an initial PNA screening, in which was assessed the ability of the R8-PNA to reduce amounts of target miRNAs. The second set of data (3 experiments), with more treatment points, was run to verify the effects of combined treatment on apoptosis induction, and intracellular miRNAs content was assessed to confirm that apoptosis induction was associated with target miRNAs inhibition. So, experiments were conducted in different times and this may explain the difference between miRNA amounts.
Point 9. Line 291-292, where is the data that supports this statement? A figure reference should be added here.
Answer. Data relative to unrelated PNA are reported on supplementary Figure S5 now cited in the main text (line 302 page 8)
Point 10. In Figure S7, the figure legend mentions C and D panels, but they are missing in the figure.
Answer. Figure S7 was emended adding panels C e D about LoVo cells
Point 11. In Figure S9, the figure legend is missing panels C and D.
Answer. You are right. Sorry for this. Legend to the figure S9 was emended adding the description of panels C and D.
Point 12. The data presented in Figure 5A and Figure S10A are inconsistent. Same cell line (HT29) and the same Annexin V assay, in Figure 5A, the SFN group has a ~ 40% increase of apoptotic cells compared to the control (untreated, -), while in Figure S10A the SFN group has a ~ 20% increase of apoptotic cells, assuming it is also comparing to the control. How can the results be off so much? Are these two figures based on two different sets of data? Why they are not combined and averaged? This part needs to be justified.
Answer. Figure 5 of the main text reported the values of % apoptotic cells. Figure S10 reported the value obtained by subtracting the SFN values to the control DMSO-treated cells. This was specified in the legend to Figure S1, by including the sentence: “The data included in panel A were obtained by subtracting the SFN values to that of control DMSO-treated cells”.
Point 13. From Figure 3 to Figure 5, there is no indication of statistical analysis or statistical significance among the treatment groups. This greatly affects the significance of the study content and the authors should include a justification in the discussion indicating the limits of the study as well as reasoning the scientific soundness of their findings.
Answer. Figures 3-5 have been amended, and statistical analysis included.
Sincerely,
Alessia Finotti
Department of Life Sciences and Biotechnology, University of Ferrara
Reviewer 3 Report
Data are interesting. However, it would be useful to discuss potential differences between the results obtained in vitro and their application in vivo for cancer treatment.
Author Response
Reviewer #3
Point 1. Data are interesting. However, it would be useful to discuss potential differences between the results obtained in vitro and their application in vivo for cancer treatment.
Answer. This is a key point and the thank the reviewer for raising this issue. We discussed the importance and feasibility pre-clinical studies using in vivo model systems by adding in the Discussion section the following sentence: “Moreover, the validation of the efficacy of co-treatments using different PNAs or the combined strategy based on antimiRNA-PNAs and sulforaphane is a required step, for proposing this approach for the possible development of tailored protocols for therapeutic interventions in the management of colon cancer patients. In vivo experiments are feasible, as demonstrated by several studies demonstrating in vivo activity of SFN [87-89]. Concerning PNAs, example of the employment of these reagents in vivo tumor models have been reported [90,91]” (page 11, Lines 432-438).
Sincerely,
Alessia Finotti
Department of Life Sciences and Biotechnology, University of Ferrara
Round 2
Reviewer 1 Report
The authors have addressed all my concerns.